# Research Advances of Beneficial Microbiota Associated with Crop Plants

**DOI:** 10.3390/ijms21051792

**Published:** 2020-03-05

**Authors:** Lei Tian, Xiaolong Lin, Jun Tian, Li Ji, Yalin Chen, Lam-Son Phan Tran, Chunjie Tian

**Affiliations:** 1Key Laboratory of Mollisols Agroecology, Northeast Institute of Geography and Agroecology, Chinese Academy of Sciences, Changchun 130102, China; tianlei@iga.ac.cn (L.T.); linxiaolong@iga.ac.cn (X.L.); jili@iga.ac.cn (L.J.);; 2China North Vehicle Research Institute, China North Industries Group Corporation Limited, Beijing 10072, China; teunj2011@126.com; 3Colleges of Resources and Environment, University of Chinese Academy of Sciences, Beijing 100049, China; 4Institute of Research and Development, Duy Tan University, 03 Quang Trung, Da Nang, Vietnam; Stress Adaptation Research Unit, RIKEN Center for Sustainable Resource Science, 1-7-22, Suehiro-cho, Tsurumi, Yokohama 230-0045, Japan

**Keywords:** endosphere, microbiota, phyllosphere, plant–microbe interaction, rhizosphere

## Abstract

Plants are associated with hundreds of thousands of microbes that are present outside on the surfaces or colonizing inside plant organs, such as leaves and roots. Plant-associated microbiota plays a vital role in regulating various biological processes and affects a wide range of traits involved in plant growth and development, as well as plant responses to adverse environmental conditions. An increasing number of studies have illustrated the important role of microbiota in crop plant growth and environmental stress resistance, which overall assists agricultural sustainability. Beneficial bacteria and fungi have been isolated and applied, which show potential applications in the improvement of agricultural technologies, as well as plant growth promotion and stress resistance, which all lead to enhanced crop yields. The symbioses of arbuscular mycorrhizal fungi, rhizobia and *Frankia* species with their host plants have been intensively studied to provide mechanistic insights into the mutual beneficial relationship of plant–microbe interactions. With the advances in second generation sequencing and omic technologies, a number of important mechanisms underlying plant–microbe interactions have been unraveled. However, the associations of microbes with their host plants are more complicated than expected, and many questions remain without proper answers. These include the influence of microbiota on the allelochemical effect caused by one plant upon another via the production of chemical compounds, or how the monoculture of crops influences their rhizosphere microbial community and diversity, which in turn affects the crop growth and responses to environmental stresses. In this review, first, we systematically illustrate the impacts of beneficial microbiota, particularly beneficial bacteria and fungi on crop plant growth and development and, then, discuss the correlations between the beneficial microbiota and their host plants. Finally, we provide some perspectives for future studies on plant–microbe interactions.

## 1. Introduction

A microbiota comprises all microbes, including viruses, bacteria, archaea, protozoa, and fungi that are present in a special environment, which is usually investigated together with its host and surrounding environment such as plant, animal gut, and soil, whereas a microbiome includes all the living microorganisms (viruses, bacteria, archaea, and lower and higher eukaryotes), their genomic sequences, and the environmental conditions surrounding the entire habitat [1,2,3]. Microbes are highly diverse and abundantly present in the Earth’s ecosystem, and can survive in extreme environments [1,2]. Microbes are closely associated with their living environment, and environmental changes can alter different microbial species [1,4,5]. Thus, different crop plant species can selectively assemble various microbes in their rhizosphere, phyllosphere, and endosphere [6,7]. Studies have revealed that the microbiota associated with plants is crucial in determining plant performance and health, because certain beneficial microbes improve plant growth and resistance to stresses [8,9,10,11,12].

Crop plants which are cultivated on a large scale in fields for human consumption, livestock feeding, profits, or as raw materials for industrial products are often referred to as agronomic crops [13,14]. With the industrialization and development of human society, improvement of plant productivity and product quality in a natural environment remains a concern. The technologies for crop cultivation have also changed with the use of machines, chemical fertilizers, and pesticides, and thus the composition of microbiota associated with plants has been affected [12,15,16]. Increasing evidence has shown that plant-associated microbiota plays important roles in plant growth and development and is able to provide protection for plants against invading pathogens and various abiotic stresses [17,18]. Moreover, studies have proven that different assemblies of microbiota occur in association with different members of the plant kingdom [19,20,21]. One recent report revealed that the nitrogen (N) absorptions by both the rice (*Oryza sativa*) indica and japonica species are associated with the *NRT1.1B* gene, encoding a rice nitrate transporter and sensor, which is responsible for root-associated microbiota composition and N usage in field-grown rice [22]. Community structures of rhizomicrobiomes associated with four traditional Chinese medicinal herbs (*Mentha haplocalyx*, *Perilla frutescens*, *Glycyrrhiza uralensis,* and *Astragalus membranaceus*) presented significantly different bacterial and fungal communities after culturing in the same soil and air conditions [20].

The advance of next generation sequencing (NGS) technologies has presented and promoted an effective method for studying the plant–microbe interactions by facilitating the investigations of whole plant-associated microbial communities [23,24,25]. Numerous research groups have focused on studies of microbial communities using the advanced sequencing technologies. For instance, using NGS technologies, Perez-Jaramillo et al. reported a link between the rhizosphere microbiota compositions of domesticated and wild common bean (*Phaseolus vulgaris*) with their genotypic and root phenotypic traits [26], while Shenton et al. performed a study to examine the effects of *O. rufipogon* (wild rice) and *O. sativa* (cultivated rice) on the rhizosphere bacterial community compositions [27].

Studies have reported that certain species of bacteria and fungi are closely associated with their host plants [28,29]. Various genera of beneficial microbes associated with crop plants have been studied for their symbiotic formation with the host plants in recent years [30,31]. Previous reports have detailed the importance of microbiota on their host plants; however, only a few reviews have been available on this important topic in a systemic manner [17,18,32]. Thus, in this review, we have made an effort to systemically illustrate the latest research advances of beneficial microbiota associated with crop plants and make suggestions for the research trends on the plant and microbiota relationships.

## 2. Beneficial Microorganisms for Crop Plants

### 2.1. Plant Growth-Promoting Bacteria

Plant growth-promoting bacteria (PGPBs) occupy a large proportion of plant microbes and are able to promote plant growth under various environmental conditions in different ways. PGPBs such as beneficial plant-symbiotic bacteria, including rhizobia and *Frankia* spp., form a symbiotic structure in plant roots [30,31] (Figure 1a). The host plants of rhizobia and *Frankia* spp. are leguminous and non-leguminous plants (e.g., actinorhizal plants), respectively [30]. The symbiosis of rhizobia with legumes is a host-specific association [33], and the rhizobia from each genus are well known to nodulate a specific host legume to generate nodules that fix atmospheric nitrogen (N_2_) [33]. Among the rhizobia, the *Rhizobium* spp. are often associated with chickpea (*Cicer arietinum*) and common bean, whereas the *Bradyrhizobium* spp. are largely found to nodulate cowpea (*Vigna unguiculata*) and soybean (*Glycine max*) plants [34]. For instance, *B. japonicum* fixes the N_2_ to promote plant growth, and thus improves soybean production and reduces the fertilizer requirement in the fields [35]. *Rhizobium*-legume association is used as a biofertilizer in several countries due to the effective of N_2_ fixation of rhizobia [36]. It is well known that inoculation of rhizobia also helps to save the N source for cereal production, thus, reducing the application of N fertilizers which in turn conserves the environment [31,37,38,39,40]. Interestingly, Wu et al. indicated that inoculation of rice with one species of rhizobia, *Sinorhizobium meliloti*, helped rice growth by elevating the expression of genes that function to enhance cell expansion and accelerate cell division [41]. Furthermore, another study revealed that the alliance of arbuscular mycorrhizal (*Glomus mosseae*) and rhizobial (*R. leguminosarum*) symbioses alleviated damage to clover (*Trifolium repens*) by root hemiparasitic *Pedicularis* species [42].

The N_2_-fixing *Frankia*, one genus belonging to the actinobacteria group, form the actinorhizal nodules with actinorhizal plants, and promote the growth and abiotic resistance for several woody plants, such as *Alnus rubra*, *Hippophae tibetana*, *Elaeagnus angustifolia,* and *Casuarina glauca* [30,43,44,45,46]. Root nodule formation in plants by *Frankia* spp. is a host plant-specific process, and the *Frankia* spp. can be clustered into three clusters that include some identified species, namely (1) *Frankia alni*, *F. casuarinae*, and *F. torreyi*; (2) *F. coriariae* and *F. datiscae*; and (3) *F. elaeagni* and *F. irregularis*, which represent the associated *Alnus*, *Dryas* and *Elaeagnus* hosts, respectively [47,48,49]. Ngom et al. determined that the *Frankia* spp. in Cluster 1 can also promote the growth of *Casuarina glauca* through symbiosis [50]. Several reports revealed that *Alnus*, *Hippophae,* and *Elaeagnus* plants can resist drought and salt stress with the help of specific *Frankia* strains [51,52]. On the basis of the results of genome analyses, *Frankia* strains possess some shared or specific genes for promoting plant growth, as well as genes related to N_2_ fixation [49]. Further studies are required to understand the underlying mechanisms, as well as the clustering of the *Frankia* species based on their specific symbiotic relationship with the host plants [52,53,54].

Other types of PGPBs, either residing in the rhizosphere or phyllosphere or endosphere, can also promote plant growth and stress resistance [5]. These PGPBs have some beneficial traits, such as the ability to produce indole-3-acetic acid (IAA), 1-aminocyclopropane-1-carboxylic acid (ACC) deaminase and siderophores, as well as to solubilize phosphate [55,56,57]. Furthermore, some PGPBs also compete with certain pathogenic bacteria or fungi for nutrients or niches, and thus inhibit the spread of pathogens and reduce damage to plants [58]. Some PGPB species of *Bacillus* and *Pseudomonas* genera have been studied for their potential applications in sustainable agriculture development [59,60,61]. For instance, PGPBs belonging to *Bacillus* and *Pseudomonas*, as well as *Azospirillum* and *Azotobacter* genera were shown to improve alkaloid contents in the medicinal plant *Withania somnifera* in a two-consecutive-year experiment [62], while *P. putida* and *P. fluorescens* were proven to improve tropane alkaloid production in *Hyoscyamus niger* plants under water scarcity [63]. In this context, Maggini et al. proposed *Echinacea purpurea* as a new model plant to study crosstalk between medicinal plants and bacterial endophytes, with the aim to discover bioactive compounds [64]. Their suggestion was made based on the finding that the secondary metabolite levels in non-infected and infected plants were different, indicating that the bacterial infection modulated the biosyntheses of several secondary metabolites [64]. Additionally, the transcript levels of a gene involved in the alkamide metabolic pathway were higher in the roots of infected *E. purpurea* than in that of the non-infected control [64]. Several secondary metabolites produced by PGPBs can also be used to control pathogens and salinity [60,65]. Several studies have also reported that PGPBs lead to induced systemic resistance (ISR) in plants, which helps the plants to resist the pathogens [66,67,68] (Figure 1b).

### 2.2. Plant Growth-Promoting Fungi

Plant growth-promoting fungi (PGPFs) have gained immense attention as biofertilizers due to their role in maintaining plant quality and quantity and their environment-friendly relationship [69] (Figure 1a). Advancements in the development of PGPFs for crop cultivation have been achieved in *Salvia miltiorrhiza*, grapevine (*Vitis vinifera*), and lettuce (*Lactuca sativa*) [69,70,71,72]. For instance, *Trichoderma* spp., *Ganoderma* spp., and yeasts (*Saccharomyces* spp.) have been used as PGPFs in several studies under both normal growth and environmental stress conditions [70,73,74,75,76,77,78,79]. For example, Jaroszuk-Scisel et al. reported that a *Trichoderma* strain, isolated from a healthy rye (*Secale cereale*) rhizosphere with the ability to produce auxin, gibberellins, and ACC deaminase in vitro, and to inhibit the growth of *Fusarium* spp., could colonize the roots of wheat (*Triticum aestivum*) seedlings and enhance stem growth [73]. Another study revealed that *T. asperellum* could induce peroxidase and polyphenol oxidase, as well as the cell wall degrading enzymes chitinase and β-1,3-glucanase in lettuce plants to defend the infected plants against the leaf spot disease [70]. Several isolated killer yeast *S. cerevisiae* strains were shown to serve as PGPFs for controlling *Colletotrichum gloeosporioides* in grape planting [80]. Cell wall isolated from yeast (*Rhodosporidium paludigenum*) could induce disease resistance against *Penicillium expansum* in pear (*Pyrus pyrifolia*) fruits by inducing the activities of defense-associated enzymes such as β-1,3-glucanase and chitinase, and by upregulating expression of pathogenesis-related (PR) genes in plants [75].

Arbuscular mycorrhizal fungi (AMFs), equal to the term ”Glomeromycota” at the fungal phylum level, can benefit plants as PGPFs during nutrient absorption and stress resistance by acting as symbionts to their hosts [81,82] (Figure 1a). Being one group of obligate endomycorrhizas that has occurred for nearly 400 million years, AMFs also play a key role in liverworts (*Marchantiophyta*), hornworts (*Ceratophyllum demersum*) and lycophytes by helping them adapt to the land during the early years [83]. Spagnoletti et al. reported that AMFs promote soybean production by helping soybean plants resist arsenic (As) contaminated soils [82], while Cely et al. indicated that inoculation of *Rhizophagus clarus* increased the soybean production under field conditions [84]. Moreover, AMFs improved rice production by reducing the total content of As, and improving the nitrogen/carbon (N:C) ratio in rice grains [85,86]. Furthermore, one study reported that selenium (Se) × AMF interactions promoted both soybean and forage grass (*Urochloa decumbens*) for Se absorption, which improved the quality of the two plants used as animal feed [87]. *Rhizoglomus intraradices*, one model species of AMFs, help rice, soybean, ginseng (*Panax ginseng*), and other crops resist drought and cold stresses and promote nutrient absorption [10,88,89]. *R. intraradices* also promoted rice plants to resist blast disease (caused by *Magnaporthe oryzae* spores), and the transcriptome data revealed that *R. intraradices*-inoculated rice plants exhibited significantly higher expression levels of genes related to the auxin and salicylic acid signaling pathways than the noninoculated rice plants in response to *M. oryzae* [11].

### 2.3. Biocontrol Agents

The biocontrol agents (BCAs), including various beneficial AMFs, PGPBs, and PGPFs have been investigated by several scientists in the last 30 years [11,90,91,92,93]. For example, various species of *Pseudomonas*, *Saccharomyces*, *Streptomyces,* and *Trichoderma* genera are the most effective BCAs that help plants acquire nutrients and control the plant pathogens [71,91,94]. These species possess a number of antagonistic mechanisms, including competition for nutrients and space with pathogenic fungi or bacteria, producing antifungal and bacterial compounds for resisting pathogenic fungi and bacteria, and direct parasitism [71,91,94]. In addition, these microbes can trigger plant resistance by strengthening the cell wall and enhancing the physiological and biochemical responses of plants to stresses [95].

## 3. Why Plants Need Beneficial Microbiota

As discussed above, a myriad of microbes populated on plant surfaces or inside the plants can considerably affect plant growth, nutrient uptake, and resistance to environmental stresses [11,90,91,92,93]. Therefore, gaining deep insights into the mechanisms underlying the evolution, compositions, and functions of plant microbiomes would open new avenues to enhance crop health and yield, ensuring global food security [96]. Wei et al. proposed a framework for plant breeding, which would enable plants to obtain economically novel phenotypes by altering plants’ genomic information along with plant-associated microbiota [97]. In the breeding strategy of plants, microbes are considered to be one of the direct targets subjected to selection for achieving a desirable plant phenotype [97].

Bai et al. reported different bacterial communities in the rhizophere and phyllosphere [98]. Roots secrete some organic C and N sources into the rhizosphere, which help in enrichment and assembly of the soil microbes in the rhizosphere [99,100,101]. A number of studies have revealed that the correlation between microbiota and related metabolic pathways in plants mutually affects the growth and responses of plants to environmental stresses [21,102,103]. Considering various compounds secreted by different plants, a different rhizosphere microbiota is associated with a different plant species to support their performance and stress resistance [20,104,105]. The beneficial microbiota, containing AMFs, PGPBs, and PGPFs, also affects the metabolism of various substances in plants [10,106,107]. This mutual relationship between microbiota and plants effectively supports plant performance and resistance to adverse environmental conditions [8,9,10,11] (Figure 1b).

Soil represents an extremely rich microbial reservoir on the Earth [108]. Soil is the origin of the rhizosphere microbiota, and a driver of microbial community formation [109,110]. The diversity and abundance of a microbiota in soil are about (1~10) × 10^9^ CFU bacteria/g soil and (1~10) × 10^5^ colony-forming unit (CFU) fungi/g soil in normal soils [111,112]. Rhizosphere is the soil surrounding the plant roots within 2 to 5 mm, in which numerous microbes assemble [25,113,114]. The rhizosphere of the crop plants acts as the major ecological environment for establishment of plant–microbe interactions. The mutual interactions involve colonization of numerous microbes growing in the inner parts or on the surface of roots, and the microbiota result in associative, symbiotic, and even parasitic collaborations among the plants and the associated microflora [115,116]. Soil, as the natural medium for bacteria and fungi, provides C and N sources needed by these microbes [117,118]; however, these C and N sources are normally provided by plants, further illustrating the mutual relationship between plants and microbes. Furthermore, the soil characteristics differ due to the flora grown on it, the microbes in the soil, and soil management by humans [19,20,119].

Whereas the use of chemicals, such as insecticides, herbicides and pesticides, hampers the quality of plant products, and thus adversely affects human health [120,121], extracts and metabolites of PGPBs and PGPFs effectively protect plants from various biotic and abiotic adversities [60,77,122]. Thus, if the microbiota containing PGPBs and PGPFs is effectively used, the problems regarding crop production could be solved in an efficient and environment-friendly manner. Endophytic microbiota residing inside the roots, stems, and leaves of plants help to maintain plant health and help plants resist adverse conditions [123]. It has been reported that plants selectively construct the community of their endophytic microbiota, which helps in upregulating expression levels of stress resistance-related genes in plants [123].

In addition, plants need soil microbiota for degrading their residues [124]. Plant residues are abundantly found in the soil [124,125]. Although chemical or physical methods can help degrade the residues, the natural degradation is carried out by soil microbes [125]. The microbiota assembles and is enriched in the plant residues and further degrades them into macro- or micromolecules, which can serve as soil organic matter [125]. For instance, the utilization of plant residues and enrichment of soil microbiota could improve the organic matter content and composition of the soils in northeast China [126,127]. Improved soil organic matter properties protect the soil from destruction and maintain its nutrient content, which would make the soil more suitable for crop production [128,129]. In addition, when plants are grown under biotic or abiotic stress conditions, such as in pathogenic fungi-contaminated, heavy metal-contaminated, or alkaline soils, the plant-associated microbiota helps them resist stresses by mediating plant abscisic acid (ABA) levels, plant jasmonic acid (JA) levels, or Bradford reactive soil protein protection [11,130,131].

## 4. Influences of the Community Compositions of Rhizosphere, Phyllosphere, and Endosphere Microbiota on Growth and Performance of Crop Plants

A few advances have been made to illustrate the diversity and community structure of the microbes associated with crop plants, and their relationship with the changes in plant metabolites and gene expression [21,102,132]. Community compositions of the rhizosphere, phyllosphere, or endosphere microbiota associated with plants indicate the health or nutrient conditions of the plants [21,115,116,133,134]. On the one hand, the highly enriched pathogenic fungi in the rhizosphere indicate stress conditions that can adversely affect the growth of crop plants [10,135]. On the other hand, the beneficial rhizospheric microbes improve plant growth, nutrient absorption, and development based on the mechanisms, such as organic matter mineralization, disease containment against soil-borne pathogens, N_2_ fixation, potassium (K) and phosphate solubilization, and IAA and ACC deaminase production [115,116]. Thus, the soil/plant/microbial relationship must be appropriately maintained for sustainable agricultural practices [115,116]. Phyllospheric microbes, consisting of mostly bacteria and fungi, can act as (i) beneficial mutualists that improve plant growth and resistance to environmental stresses; (ii) commensals that use the leaf habitat for their own growth, development, and reproduction; or (iii) antagonistic pathogens [136,137,138] (Figure 1a). A study of 14 phylogenetically diverse plant species grown under controlled greenhouse conditions showed high presentation of *Bacillus* and *Stenotrophomonas* genera in their phyllosphere microbial communities, which showed antagonistic potential toward *Botrytis cinerea* [134], a foliar pathogen that causes gray mold disease in more than 200 dicot crop plants [139].

Endophytic bacteria and fungi, inhabiting within the plant tissues, play crucial roles in plant growth, fitness, development and protection without causing any evident damage to the host plants [140,141]. These endophytic bacteria and fungi reside on intercellular spaces in the plants for a certain period of their life cycle, and obtain carbohydrates, amino acids, and inorganic nutrients from their hosts [142]. Some endophytic bacteria or fungi also produce IAA, soluble phosphate, and siderophores, which can promote the host plant growth [143]. For instance, Borah et al. studied the endophytic microbes of cultivated rice (*O. sativa*) and wild rice (*O. rufipogon*) plants and reported their diversity in terms of their functional characteristics related to multiple traits associated with promotion of plant growth and development [144]. Murphy et al. screened fungal endophytes from wild barley (*Hordeum murinum*) for testing the beneficial and promoting traits in cultivated barley (*H. vulgare*) and found that some fungi could promote the growth of cultivated barley [145].

Furthermore, a few studies have reported that both artificial management of plant growth environment and conditions, and plant domestication affect the community structure of rhizosphere and endosphere microbiomes [27,119,146,147]. Wild plant species preserve more diversity in associated microbiota for their survival against stress resistance than the domesticated crops. A recent study reported that common wild rice (*O. rufipogon*) grown in wildland of lower soil nutrients showed higher rhizobacterial diversity than the cultivated rice grown under cultivated field conditions with higher soil nutrient [28]. Martín-Robles et al. discussed the impacts of domestication on the arbuscular mycorrhizal symbioses of 14 crop species under different available phosphorus conditions, and reported that in response to increased available phosphorus, domestication decreased the AMF colonization in domesticated plants as compared with their wild progenitors [148]. Thus, more studies revealing the influence of plant domestication on the compositions of microbiomes are warranted, and the rhizosphere microbiota in wild and cultivated species grown in the field conditions are essentially required to study the correlation of microbiota and host plants.

A plant can release various chemical compounds into the environment that impose either direct or indirect allelochemical effect on another plant [149] (Figure 1b). In addition, studies have revealed that the production of allelochemicals by plants significantly alters the structure of associated microbiomes [150,151,152]. However, the influence of microbiota on the allelochemical production in plants remains unclear. Moreover, the relationships of specific allelochemicals with a particular microbiota remain unillustrated. A recent study by Li et al. concluded that replantation of Sanqi ginseng (*P. notoginseng*) is not recommended mainly due to the accumulation of soil-borne pathogens and allelochemicals (e.g., ginsenosides) [153]. Li et al. also demonstrated that reductive soil disinfestation approach could effectively alleviate the replantation failure of Sanqi ginseng with allelochemical degradation and pathogen suppression [153] (Table 1). However, the reason behind the continuous cropping obstacle on the community compositions of microbiomes remains the task of future research.

## 5. Applications of Individual Microbes for Improvement of Crop Performance and Soil Ameliorations

Various beneficial microorganisms have been commercialized [154]; however, their efficacy has not always been consistent in terms of benefiting crop plants in fields. Nevertheless, combining beneficial microbes for application to crop plants has been shown to be more effective in improving plant performance. The PGPBs and PGPFs have been applied in agricultural practices for 30 years. For instance, applications of the AMFs have been known to improve the yield of cotton (*Gossypium hirsutum*) and soybean, as well as soybean resistance to the *Macrophomina phaseolina* pathogen and stress associated with As [82,84]. AMFs also promote N transfer from soil to plants as indicated by a study of Tian et al., in carrot (*Daucus carota*) [155]. The authors also reported that *R.*
*intraradices*-mediated N transfer successively induced the expression of N metabolism-related genes in the intraradical and extraradical mycelia, as well as in the host plants [155]. The interactive effects of the AMF *R. irregularis* and the PGPB *P. putida* synergistically improved the growth and defence of the wheat host plants against pathogens [156]. With the global warming, drought has been one of the major limiting conditions for crop production [157]. Studies have shown that AMFs enhanced rice and wheat in resisting drought by improving the plant physiological index [158,159]. Application of PGPBs such as *Azospirillum*, *Azotobacter*, *Pseudomonas,* and *Bacillus* species improved the alkaloid contents in two medicinal plants *W. somnifera* and *H. niger* under limited water conditions [62,63]. With regards to the endosymbiotic PGPBs, Chandrakala et al. screened and isolated one *Rhizobium* species from rice rhizosphere, which could solubilize silicate and promote plant growth potential of the rice plants [160]. *B. japonicum* has been shown to help soybean to resist a high content of As in the soil [161], whereas *S. fredii* could effectively nodulate numerous different legumes, thereby promoting growth of the host plants [162]. Vemulapally et al. screened and isolated *Frankia* spp. from the nodules of *Casuarina* spp., one of which (*F.*
*casuarinae* CcI3 strain) could form nodules in roots of *C.*
*equisetifolia* seedlings after its inoculation to the plants and promote plant growth [163]. Forchetti et al. demonstrated that certain bacterial species can produce JA and ABA [4]. However, whether the contents of these hormones in plants can be influenced directly by the associated microbes remains unclear. Nevertheless, hormone-producing bacteria or fungi are not always beneficial; for example, *Gibberella fujikuroi* can produce gibberellin and cause rice bakanae disease [164] (Table 1). Thus, studies on the mutual effects of microbes and host plants on hormone metabolism in both plants and microbes in the context of plant and microbiota relationships have been novel and interesting, and thus this topic would deserve more attention from the research community.

Microbe-mediated remediation of environmental contaminants requires more research for heavy metal-polluted soils and saline-sodic soils **(**Figure 1b). With the industrialization, environmental contaminants, such as heavy metal, household garbage, and excessive N and phosphate in the soil or water, have seriously affected the quality of humans’ and plants’ life. Mitigation of heavy metal contamination in soil by fungal bioremediation has proceeded for several years [165,166], and studies for exploring heavy metal-polluted soils for agriculture by cultivating stress-tolerant plants with the help of microbes are also in process [167]. Cornejo et al. revealed that AMFs assisted *Oenothera picensis* in resisting copper stress and helped the bioremediation of copper-polluted soil [130]. Saline-sodic soils have been studied for several years to improve their soil characteristics for crop cultivation with higher productivity [19,119]. Chemical and biological ameliorations of saline-sodic soils are the most effective ways; however, both methods affect the compositions of soil microbiomes. For example, Luo et al. revealed that ameliorations of saline-sodic soils by using chemicals or plant planting affected the soil microbiota [19,168], while Shi et al. found various responses of microbial communities and enzyme activities to various chemical and biological amendments in the saline-alkaline soils [119] (Table 1). Research on plant-associated microbiota to improve the soil characteristics is further warranted to facilitate microbiota applications in different aspects of agriculture and life, and more studies considering microbiota applications along with plant cultivation for soil ameliorations are required.

As discussed earlier, beneficial microbes can also improve the useful secondary metabolites in medicinal plants, demonstrating the wide application potential of microbes and microbiota to various types of crops. Application of a microbial consortium of various PBGBs, including *Azospirillum*, *Azotobacter*, *Pseudomonas,* and *Bacillus* spp., improved the levels of alkaloids, which have pharmacological properties [62,63], in *W. somnifera* and *H. niger* [62,63]. Furthermore, Huang et al. advocated *S. miltiorrhiza*, which is a herb widely used in traditional Chinese medicine, as a model species for investigating how different microbes can interact with medicinal plants to alter the production of phytochemical compounds [169]. Additionally, it is well known that *Taxus chinensis* can produce taxol, an anticancer metabolite [170], and its endophytic fungi *Paraconiothyrium* spp. can also produce the same anticancer metabolite [171,172,173]. However, it is not clear whether the content of taxol in *T. chinensis* could be improved by its endophytic fungi, as well as the relationship of *T. chinensis* and associated microbiota in terms of metabolite production has also not yet been determined. Thus, the effect of microbiota on the metabolic pathways of their host plant deserves more studies in future (Figure 1b).

## 6. Conclusions and Future Perspectives

Microbiota plays important roles in plant growth, development, productivity, and resistance to environmental stresses. An increasing body of research has been focusing on the associations and relationships between microbiota and host plants. Although some progresses have been made in recent years, applications of microbes and microbiota to improve crop productivity still require more research. Both theoretical research and applied research of microbes and microbiota in connection with plants remain a major task for the scientific community. In the future, AMFs, PGPBs, and PGPF should be applied and utilized more effectively in agriculture. The signaling pathways of AMFs, and rhizobia and *Frankia* spp. with their host plants have been studied for a long time but are still under debate in plant science. Various genes in microbes related to hormones such as strigolactones and nutrient uptake similar to nitrogen uptake should be studied to uncover the mutual connection with the host plants.

In the future, microbiota research using both basic and applied research approaches in connection with crop plants should be focused on the following: (1) Mutualistic symbioses of the fungal and bacterial microbes with the host plants; (2) plant diseases caused by the environmental microbiota; (3) mechanistic studies and explorations of PGPBs and PGPFs for the improvement of plant growth, productivity and resistance, and yield quality; (4) allelochemical effects between plants and microbiota; and (5) mechanisms underlying the degradation of plant residues by the microbiota for acceleration of practical use in this field. Future research on plant and microbiota associations and relationships could play an essential role in ensuring food security in response to global climate change.

## Figures and Tables

**Figure 1 ijms-21-01792-f001:**
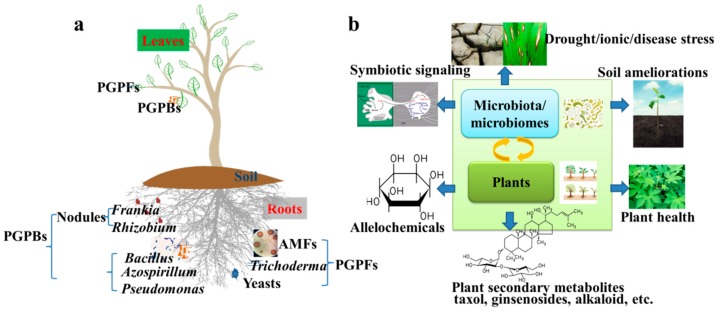
(**a**) Beneficial microbes and microbiota for crop plants. Beneficial microbiota assembles in rhizosphere, phyllosphere, and endosphere. Arbuscular mycorrhizal fungi (AMFs), *Frankia* spp. and *Rhizobium* spp. act as symbionts for plant roots; (**b**) Future research trends of microbiota and microbiomes associated with crop plants. PGPBs, plant growth-promoting bacteria; PGPFs, plant growth-promoting fungi.

**Table 1 ijms-21-01792-t001:** Several representative applications of microbes in crop plants. ACC, 1-aminocyclopropane-1-carboxylic acid; AMFs, arbuscular mycorrhizal fungi; IAA, indole-3-acetic acid; JA, jasmonic acid; ABA, abscisic acid; PGPBs, plant growth-promoting bacteria; PGPFs, plant growth-promoting fungi.

Research Aspects	Microbes/ Microbiomes	Beneficial Traits	References
Rhizobia/ *Frankia* in colonization with host plants	Rhizobia/ *Frankia* spp.	*Rhizobium* spp. promoted the growth of rice seedlings and solubilized silicate;	[160]
		*Bradyrhizobium japonicum* helped soybean in resisting arsenic contamination in soil	[161]
		*Frankia casuarinae* CcI3 formed nodules and promoted the growth of *Casuarina equisetifolia*	[163]
Positive effects of mutualistic symbioses with fungi on stress resistance	AMFs	*Rhizoglomus intraradices* reduced the relative abundance of pathogenic fungi in plant soil	[10]
		AMFs promote rice and wheat in resisting drought	[158,159]
PGPB/PGPF applications in improving plant growth, productivity, and resistance	PGPBs	Productions of IAA, ACC deaminases, JA and ABA	[55,56,57]
	PGPFs	*Trichoderma* spp. and yeasts help plant in resisting diseases	[70,73,74,75,76,77,78,79]
Allelochemical effects/soil ameliorations	Microbiomes in soil	Reductive soil disinfestation can reduce the allelochemical effect in soil.	[153]
		Soil amendments associated with changes in the compositions of soil microbiomes	[19,119]
Effects of microbiota on the production of secondary metabolites in plants	PGPBs/AMFs/*Paraconiothyr-ium*	AMFs improve the content of gesenosides in *P. ginseng* planting	[10]
		Endophytic fungi *Paraconiothyrium* spp. in *Taxus chinensis* can produce taxol	[171,172,173]
		*Azospirillum*, *Azotobacter*, *Pseudomonas* and *Bacillus* improved the alkaloid content in *Withania somnifera*	[62]
		*P. putida* and *P. fluorescens* were proved to improve tropane alkaloid production of *Hyoscyamus niger* under water deficit stress	[63]

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
