# Peer review of "Research Advances of Beneficial Microbiota Associated with Crop Plants"

_ijms, 2020, doi:10.3390/ijms21051792_

Round 1

Reviewer 1 Report

The authors have addressed all required issues and improved a scientific quality of the manuscript.

Author Response

Response: Thank you very much for supporting our manuscript. Once again, we very highly appreciate your kind comments and suggestions that have greatly helped us improve the quality of our manuscript.

Reviewer 2 Report

The review manuscript "Research Advances of Beneficial Microbiota Associated with Crop Plants" provides a fundamental overview of beneficial microbes for crops. The description reads fine but some sentence may be revised to be less exaggerated. 

line 46. Please be specific for what is "easy" variation? how "easy" is considered easy? and the same problem for what is "strong" adaptation, and how "strong" is strong?

line 83-84. There are plenty of review articles regarding beneficial microbes, plant growth-promoting microbes, and microbiome. Saying "no review" will be too arbitrary.

line 157-158. Tons of research and review articles have been published for ISR. Saying "a few studies" here is being to arbitrary......

In addition to the minor descriptive revisions, I would suggest the authors consider the appropriate use of terminology on microbiome study (please refer to Marchesi and Ravel. 2015. Microbiome). The definition of microbiota and micorbiome that the authors gave in the line 41-44 does not seem to be the most common use. But it is open if the authors can provide justifications.

Author Response

Reviewer #2

The review manuscript "Research Advances of Beneficial Microbiota Associated with Crop Plants" provides a fundamental overview of beneficial microbes for crops. The description reads fine but some sentence may be revised to be less exaggerated. 

Response: Thank you so much for your encouraging evaluation of our manuscript. We have revised the manuscript following your suggestions and comments to improve its quality. Please see below our responses itemized to your comments and suggestions. We hope that our revised version would now meet your expectation.

Point 1: line 46. Please be specific for what is "easy" variation? how "easy" is considered easy? and the same problem for what is "strong" adaptation, and how "strong" is strong?

Response 1: Thank you very much for this comment, with which we totally agree. During the revision, we carefully thought that this sentence did not fit to the manuscript in the whole context. Thus, we have deleted it from the revised version.

Point 2: line 83-84. There are plenty of review articles regarding beneficial microbes, plant growth-promoting microbes, and microbiome. Saying "no review" will be too arbitrary.

Response 2: Thank you very much for this comment. We have altered this sentence as below to meet the Reviewer’s comment:

“Previous reports have detailed the importance of the microbiota on their host plants; however, only a few reviews have been available on this important topic in a systemic manner [17, 18, 32].” P2, L41-43

Point 3: line 157-158. Tons of research and review articles have been published for ISR. Saying "a few studies" here is being to arbitrary......

Response 3: Thank you very much for this comment. We have slightly altered this sentence as below to meet the Reviewer’s comment:

“A number of studies also reported that PGPBs can lead to induced systemic resistance (ISR) in plants, which helps the plants to resist the pathogens [66-68] (Figure 1b).” P4, L23-25

Point 4: In addition to the minor descriptive revisions, I would suggest the authors consider the appropriate use of terminology on microbiome study (please refer to Marchesi and Ravel. 2015. Microbiome). The definition of microbiota and micorbiome that the authors gave in the line 41-44 does not seem to be the most common use. But it is open if the authors can provide justifications.

Response 4: Thank you very much for this comment. We agreed with the definition of the term “microbiome” in Marchesi et al. (2015). Thus, following the Reviewer’ advice, we have revised the sentence as below:

“A microbiota comprises all microbes, including viruses, bacteria, archaea, protozoa and fungi that are present in a special environment, which is usually investigated together with its host and surrounding environment like plant, animal gut and soil, while a microbiome includes all the living microorganisms (viruses, bacteria, archaea, and lower and higher eukaryotes), their genomic sequences, and the environmental conditions surrounding the entire habitat [1-3].” P1, L43-44 and P2, L1-4